# Role of feasibility and pilot studies in randomised controlled trials: a cross-sectional study

Amanda Jane Blatch-Jones,[1] Wei Pek,[2] Emma Kirkpatrick,[3] Martin Ashton-Key[1]

[1]National Institute for Health Research Evaluation, Trials and Studies Coordinating Centre (NETSCC), University of Southampton, Southampton, UK
[2]Faculty of Medicine, University of Southampton, Southampton, UK
[3]Southampton Clinical Trials Unit, University of Southampton, Southampton, UK

**Correspondence to**
Amanda Jane Blatch-Jones;
ajy5@soton.ac.uk

## ABSTRACT

**Objectives** To assess the value of pilot and feasibility studies to randomised controlled trials (RCTs) funded by the National Institute for Health Research (NIHR) Health Technology Assessment (HTA) programme. To explore the methodological components of pilot/feasibility studies and how they inform full RCTs.

**Study design** Cross-sectional study.

**Setting** Both groups included NIHR HTA programme funded studies in the period 1 January 2010–31 December 2014 (decision date). Group 1: stand-alone pilot/feasibility studies published in the HTA Journal or accepted for publication. Group 2: all funded RCT applications funded by the HTA programme, including reference to an internal and/or external pilot/feasibility study. The methodological components were assessed using an adapted framework from a previous study.

**Main outcome measures** The proportion of stand-alone pilot and feasibility studies which recommended proceeding to full trial and what study elements were assessed. The proportion of 'HTA funded' trials which used internal and external pilot and feasibility studies to inform the design of the trial.

**Results** Group 1 identified 15 stand-alone pilot/feasibility studies. Study elements most commonly assessed were *testing recruitment* (100% in both groups), *feasibility* (83%, 100%) and *suggestions for further study/investigation* (83%, 100%). Group 2 identified 161 'HTA funded' applications: 59 cited an external pilot/feasibility study where *testing recruitment* (50%, 73%) and *feasibility* (42%, 73%) were the most commonly reported study elements: 92 reported an internal pilot/feasibility study where *testing recruitment* (93%, 100%) and *feasibility* (44%, 92%) were the most common study elements reported.

**Conclusions** 'HTA funded' research which includes pilot and feasibility studies assesses a variety of study elements. Pilot and feasibility studies serve an important role when determining the most appropriate trial design. However, how they are reported and in what context requires caution when interpreting the findings and delivering a definitive trial.

## INTRODUCTION

Pilot and feasibility studies have an important role to play in the development of randomised controlled trials (RCTs). If appropriately used, pilot and feasibility studies can provide

**Strengths and limitations of this study**

► This paper assesses the role of pilot and feasibility studies funded by the National Institute for Health Research (NIHR) Health Technology Assessment programme.
► The study found that pilot and feasibility studies share common elements when contributing to the design of a trial.
► The study contributes to the growing literature in this area and demonstrates the value of pilot and feasibility studies to the progression to full randomised controlled trials.
► Although the data cover a 5-year period, the number of eligible studies is small and only reports from one NIHR programme.

sufficient methodological evidence about the design, planning and justification of a trial. They are often undertaken to inform elements of the main trial design, but they can also be used to reduce or eliminate problems that limit the successful delivery of trials. In 2009, the *Lancet* published a paper that highlighted the extent to which research is wasted, and that loss is as much as 85% of research investment.[1] Given the cost and time of investment from researchers and major health research funders, there is now a growing demand to assess and examine where improvements need to be made to the design and conduct of trials.[2] Poorly designed trials could include non-reference to a pre-existing systematic literature review or bias generated by inadequate concealment of treatment allocation.[1] Research by Cooper *et al* has also shown variability between external pilots and the prediction for randomisation and attrition rates.[3] As a result, much attention has primarily focused on the design, conduct and analysis of clinical research to determine where improvements are needed to reduce waste in research.

Over the last 10 years, we have seen how pilot and feasibility studies have become an important feature in terms of gathering evidence to inform the development of a full

trial. There is now an extension to the Consolidated Standards of Reporting Trials which provides guidance for pilot and feasibility studies being conducted prior to a main trial.[4] Conducting a pilot or feasibility study to determine any uncertainties prior to the main trial may help to eradicate issues and thus inform the definitive trial. More importantly perhaps is the role pilot and feasibility studies can have in modifying the design and conduct, and therefore increasing the value of the research, helping to avoid methodological design flaws and reducing the burden of research waste.

Despite the growing importance of pilot and feasibility studies, there is still a lack of clarity about the use of the two terms.[5–7] In 2008, the Medical Research Council (MRC) published guidance on developing and evaluating complex interventions to demonstrate the value and importance of pilot and feasibility studies as a key element in the development and evaluation process. However, the guidance did not attempt to explain or provide any definition for the terms 'pilot' and 'feasibility'.[7] It was not until 2 years later that Thabane *et al* reviewed the key aspects of pilot studies and provided a detailed account of pilot studies which included a number of definitions.[6] Around the same time (2009), the National Institute for Health Research, Evaluation, Trials and Studies Coordinating Centre (NETSCC) published a support document detailing what feasibility and pilot studies are.

> Feasibility studies are defined as 'pieces of research done before a main study in order to answer the question "Can this study be done?". They are used to estimate important parameters that are needed to design the main study … feasibility studies do not evaluate the outcome of interest.'

> Pilot studies are defined as 'a version of the main study that is run in miniature to test whether the components of the main study can all work together. It is focused on the processes of the main study … it will therefore resemble the main study in many respects'.[8]

These definitions have gone some way to aid the understanding of when it is appropriate to do pilot or feasibility studies as part of the definitive trial. These definitions are now widely used across the National Institute for Health Research (NIHR).

Despite the importance of the role of pilot and feasibility studies in informing RCTs, there is little empirical evidence about the use of these studies in informing future trials. For example, the *Lancet* series in 2014 did not make reference to the usefulness of pilot and feasibility studies in the context of increasing value and reducing waste in research design, conduct or analysis.[9] Lancaster *et al* and Arain *et al* provided a methodological framework to assess how pilot studies are used to inform the conduct and reporting of pilot studies.[5 10] Both described the challenges and complexities in the reporting of pilot studies. Arain *et al* further explored these complexities in relation to feasibility studies and full trials. More recently, research has begun to explore the differences between internal and external pilot studies and their contribution to main trials, and the appropriateness of pilot and feasibility studies for estimating the sample size.[3 11 12]

The aim of this study is to contribute evidence to this important gap in the current literature. The objective of this paper is to describe the process and results of how, and in what way, pilot and feasibility studies have been used to inform full RCTs.

## METHODS
The Health Technology Assessment (HTA) programme has a long history of commissioning pilot and feasibility studies. Therefore, the published reports (NIHR HTA Journal) of stand-alone pilot and feasibility studies were examined to determine which elements of research design are most often assessed. Applications for funded HTA trials were also assessed to establish how full trials were informed by previously completed pilot/feasibility studies as well as pilot studies embedded within the trial.

### Data source
An assessment of the NIHR HTA Programme over a 5-year period (2010–2014) was conducted using two retrospective groups. There were two groups due to the data being homogenous (data for group 1 were taken from the published HTA journal article, and data for group 2 were taken from the HTA application form).

In order to identify the included studies for both groups, we
1. Reviewed the project title in the application form and the journal article title.
2. Reviewed the abstract/executive summary.
3. Reviewed the full journal article or HTA application form.

### Sample selection
#### Group 1: stand-alone pilot and feasibility studies
Stand-alone pilot/feasibility studies funded by the HTA programme with a fund decision date from 1 January 2010 to 31 December 2014, which have published in the HTA Journal or are currently being prepared for publication and have been signed off by the editors (only those in production) were included. The published journal/approved final version of the published report was used as the source for data extraction. The stand-alone studies were categorised into 'pilot study', 'feasibility study' or 'both'.

#### Group 2: RCTs
Trials funded through the HTA programme with a fund decision date between 1 January 2010 and 31 December 2014 were included. The application form of a funded trial was used as the source for data extraction. The trials were categorised based on the type of pilot and/or feasibility: 'external/previous pilot study', 'external/previous feasibility study', 'internal pilot study', 'internal feasibility study' or 'other (mixed study)'.

 Blatch-Jones AJ, *et al*. BMJ Open 2018;8:e022233. doi:10.1136/bmjopen-2018-022233

## Box 1 Elements of a study design adapted from Arain *et al*

The methodological components included as reported by and included from Arain *et al*

**Methods related**
► Testing recruitment
► Determining the sample size/numbers available
► Follow-up/dropout
► Hypothesis testing
► Resources
► Randomisation
► Blinding
► Outcome measures
► Control group
► Data collection
► Further study suggested

**Intervention related**
► Dose/efficacy/safety
► Clinical outcomes
► Acceptability
► Feasibility

**In addition to the above, group 2 included**
► Delivery of the intervention
► Testing/developing materials

NIHR Evaluation, Trials and Studies Management Information System was used to identify the two groups and extract the relevant documents needed for data extraction. Search terms were used to search for relevant data to ensure the feasibility of future replications. The key search terms used were pilot, feasibility, preliminary work and earlier/previous study.

In addition to the search using key terms, a targeted search was carried out on specific areas of the application form. Focusing on specific areas of the application was relevant in identifying where the elements of the study design would most likely be described in relation to the pilot and/or feasibility study.

### Piloting
Data extraction tables for groups 1 and 2 were piloted with an initial sample of 10 studies. No changes were required to the classification system previously adopted by Arain *et al* as a result of the pilot work.

### Classification systems
The definition of pilot and feasibility studies agreed by NIHR for four programmes was used for the purpose of this study.[8] These definitions were also used by Arain *et al*.[10]

The classification systems developed by Arain *et al* and Bugge *et al* were adapted to determine what elements of a study design were assessed or used to inform the full trial (see box 1).[10 13] In both groups, the elements of the study design were examined in terms of

a. Did the study explicitly state it assessed any of these elements? (Yes/no)
b. Were there any recommended changes as a result of the assessment? A yes response was defined as the authors reported a change/recommendation to be considered but did not necessarily report what that change was. If the authors did not explicitly state a recommendation, it was assumed that no changes were required.

The text pertaining to the pilot and/or feasibility study was also extracted for quality assurance purposes.

Two additional study elements were included in group 2 which were not reported in group 1. These were '*delivery of intervention*' and '*testing/developing materials*'.

### Patient and public involvement
There was no patient or public involvement in the design of the study due to the nature of the project (part of a University of Southampton, Faculty of Medicine, BM5 Medicine, 4th Year project). There was no participant recruitment involved in the project as all data were taken from the published article or the HTA application.

### Data quality and assurance
Our approach to quality assurance was guided by the STrengthening the Reporting of OBservational studies in Epidemiology (STROBE), which, although designed for observational studies, could be applied to the processes we used in this current study.

For both groups, WP extracted all data and a second person assessed and reviewed the data to ensure the accuracy of data extraction. All of group 1 was assessed followed by 15% of group 2 (purposive sampling of 5% of the group followed by 10% randomly selected application forms). The remaining 85% was subsequently reviewed by MAK to determine the reliability and validity of the data extraction and usability of the adapted template. All disagreements were discussed by the team and were resolved by consensus. Data management was undertaken by WP with support from ABJ.

### Data analysis
Data for each study, based on the framework developed by Arain *et al* (see box 1), were captured using Microsoft Access 2010 (Microsoft Corporation, Redmond, Washington, USA).[10] The study design elements were entered onto an Access form and where a study element was reported a 'yes' response was captured. A separate Access form was developed for each included study for both groups. Both groups were exported into Microsoft Excel 2010 (Microsoft Corporation) and then subsequently into Statistical Product and Service Solutions V.22 (IMB Corporation). Excel was used to calculate the median and the range for group 1 only. Data were analysed and interpreted using descriptive statistics to determine the frequency of the study design elements and how often changes were recommended for full trials.

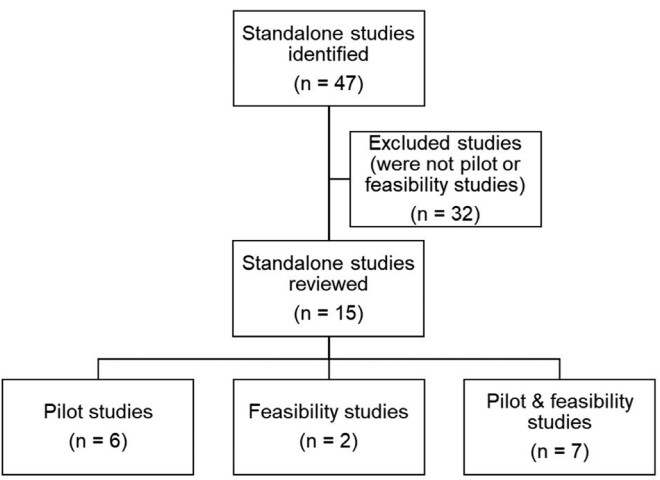

**Figure 1** The number of studies identified, excluded and categorised for cohort 1.

## RESULTS

In group 1, we identified 47 published stand-alone pilot and/or feasibility studies and in group 2 we identified 303 'HTA funded' RCTs during the 5-year period (1 January 2010–31 December 2014). Fifteen stand-alone studies were identified as eligible for group 1 and 161 funded HTA applications were identified and eligible for group 2.

### Group 1

A total of 47 stand-alone studies were identified. Thirty-two were excluded on further examination due to not being a pilot or feasibility study (we did not categorise the excluded studies by study design). The remaining 15 studies were categorised into three separate groups (see figure 1). We found that 13 of the 15 study elements included in the adapted framework were assessed in stand-alone pilot studies compared with nine study elements in feasibility studies.

In this group, it was found that seven studies used the terms 'pilot' and 'feasibility' interchangeably and it was difficult to determine, even with the NIHR definition, what type of study was undertaken. Therefore, it was not possible to accurately determine which study elements belonged to which, and in some cases the authors described the conduct of both pilot and feasibility work. The team agreed to combine pilot and feasibility together in this instance, which was also found in Arain *et al.*

The median number of participants for the stand-alone studies (n=15) was 46. Of the 15 eligible stand-alone pilot and/or feasibility studies, the most commonly reported study design element was *testing recruitment*. In all three groups (pilot studies, feasibility studies and pilot/feasibility studies), all 15 studies assessed recruitment (6/6, 2/2 and 7/7, respectively) (see table 1). Half of these also reported recommended changes to recruitment (3/6, 1/2 and 3/7, respectively). Interestingly, both feasibility studies only (2/2) and pilot/feasibility groups (7/7) assessed the need for *further study* and suggested recommended changes (*further study* referred to whether

further investigation was required using a large RCT and where future trial data could be of benefit).

### Group 2

A total of 303 'HTA funded' applications were identified. Eighty-two were excluded on examination as they were not RCTs (eg, cohort studies, diagnostic accuracy test studies and we did not categorise the excluded studies by study design) and a further 60 applications were excluded due to not being informed by any external or internal pilot and/or feasibility study. The remaining 161 applications were reviewed and subsequently grouped into five categories (see figure 2).

1. External pilot studies (n=48).
2. External feasibility studies (n=11).
3. Internal pilot studies (n=80).
4. Internal feasibility studies (n=12).
5. Other (n=10).

As the HTA application was used as the source of data extraction, the outcome of the internal pilot/feasibility study was not available (n=92). For the 59 applications where an external pilot/feasibility study was referenced, we found that not all of these studies provided information relating to the number of participants that took part in the pilot/feasibility study. We did not go back to the original journal article to retrieve this information. Therefore, it was not appropriate to estimate the median or IQR for this group.

The *others* group comprised applications that were informed by a combination of more than one preliminary study (eg, internal and/or external pilot study and/or feasibility study). Of those 10 applications,

► 6 of the 10 were informed by external pilot studies.
► 7 of the 10 were informed by external feasibility studies.
► 7 of the 10 were informed by internal pilot studies.
► 1 of the 10 was informed by an internal feasibility study.

No further analysis was conducted on these 10 applications due to the diverse nature of the study types in this subgroup.

### External pilot and feasibility studies

Of the 161 applications, 29.8% (48/161) reported or cited a previous external pilot study not recently done by the applicant and 6.8% (11/161) reported an external feasibility study. For this subset, all of the study elements (n=17) were assessed by external pilot studies but no single study assessed all 17 elements. By comparison, 13 of the 17 study elements were assessed by external feasibility studies (see table 2).

In terms of the study elements, *testing recruitment, determining the sample size and numbers available,* and the *feasibility* were the most commonly reported in both external pilot and feasibility studies. The number of reported recommended changes based on the results of the external pilot or feasibility study was however minimal. Although in some applications it was possible to detect a change, the

**Table 1** Group 1: study elements captured in pilot studies, feasibility studies and pilot/feasibility studies

| Study elements | Pilot studies only (n=6) | | | Feasibility studies only (n = 2) | | | Pilot/feasibility studies (n = 7) | | |
|---|---|---|---|---|---|---|---|---|---|
| | Assessed (A): n (%) | Recommended changes (RC): n (%) | A and RC: n (%) | Assessed (A): n (%) | Recommended changes (RC): n (%) | A and RC: n (%) | Assessed (A): n (%) | Recommended changes (RC): n (%) | A and RC: n (%) |
| Testing recruitment | 6 (100.0) | 3 (50.0) | 3 (50.0) | 2 (100.0) | 1 (50.0) | 1 (50.0) | 7 (100.0) | 3 (42.9) | 3 (42.9) |
| Determining sample size and/or number available | 5 (83.3) | 1 (16.6) | 0 | 1 (50.0) | 2 (100.0) | 1 (50.0) | 5 (71.4) | 1 (14.3) | 1 (14.3) |
| Follow-up/dropout | 4 (66.6) | 3 (50.0) | 3 (50.0) | 0 | 1 (50.0) | 0 | 5 (71.4) | 2 (28.6) | 1 (14.3) |
| Hypothesis testing | 2 (33.3) | 0 | 0 | 1 (50.0) | 0 | 0 | 1 (14.3) | 0 | 0 |
| Resources | 4 (66.6) | 3 (50.0) | 2 (33.3) | 2 (100.0) | 0 | 0 | 5 (71.4) | 1 (14.3) | 1 (14.3) |
| Randomisation | 4 (66.6) | 0 | 0 | 0 | 0 | 0 | 6 (85.7) | 1 (14.3) | 1 (14.3) |
| Blinding | 0 | 0 | 0 | 0 | 0 | 0 | 2 (28.6) | 0 | 0 |
| Outcome measures | 5 (83.3) | 4 (66.6) | 4 (66.6) | 2 (100.0) | 1 (50.0) | 1 (50.0) | 6 (85.7) | 4 (57.1) | 3 (42.9) |
| Control group | 0 | 0 | 0 | 0 | 0 | 0 | 1 (14.3) | 0 | 0 |
| Data collection | 3 (50.0) | 0 | 0 | 0 | 1 (50.0) | 0 | 3 (42.9) | 1 (14.3) | 0 |
| Clinical outcomes | 3 (50.0) | 2 (33.3) | 1 (16.6) | 1 (50.0) | 0 | 0 | 3 (42.9) | 0 | 0 |
| Dose/efficacy/safety | 2 (33.3) | 1 (16.6) | 1 (16.6) | 0 | 0 | 0 | 2 (28.6) | 0 | 0 |
| Acceptability | 4 (66.6) | 0 | 0 | 1 (50.0) | 0 | 0 | 6 (85.7) | 0 | 0 |
| Feasibility | 5 (83.3) | 0 | 0 | 2 (100.0) | 0 | 0 | 7 (100.0) | 0 | 0 |
| Suggests further study | 5 (83.3) | 4 (66.6) | 4 (66.6) | 2 (100.0) | 2 (100.0) | 2 (100.0) | 7 (100.0) | 7 (100.0) | 7 (100.0) |
| Median number of participants (IQR) [range] | 47.5 (39.25–85) [21–99] | | | 14 (7–21) [0–28] | | | 58 (35.5–173) [29–313] | | |

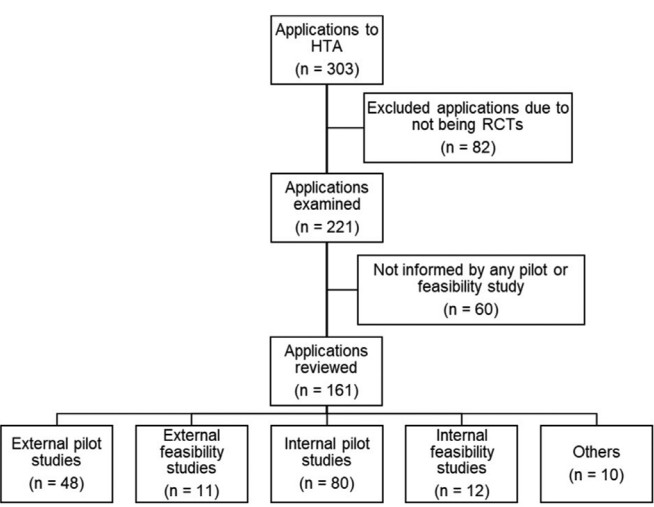

**Figure 2** Flow chart showing the number of Health Technology Assessment (HTA) funded applications for cohort 2.

authors did not explicitly state a recommended change. Therefore, it was not possible to determine whether this was based on the pilot or feasibility study, or some other factor.

### Internal pilot and feasibility studies

Of the 161 applications, 49.7% (80/161) reported an internal pilot study and 7.5% (12/161) reported an internal feasibility study. Due to the source of data extraction (the application form), it was not possible to

determine whether the funded internal pilot or feasibility study had made any recommended changes as the internal study had not yet been conducted.

For the internal studies, we found 14 of the 17 study elements were being assessed by internal pilot studies compared with 10 study elements in feasibility studies. Based on assessment only, the most common study element to be reported was *testing recruitment* (74/80 and 12/12, respectively) and *feasibility* (35/80 and 11/12, respectively) for both internal pilot and feasibility study (see table 3). There were several similarities between a number of study elements assessed by both pilot and feasibility studies.

### DISCUSSION

This study found that pilot and feasibility studies do play a role in the development and design of definitive RCTs. In both groups, it was clear that two study elements were commonly assessed: t*esting recruitment* and *feasibility*. This has important implications for the success of a trial, given that many trials struggle with recruitment and often request extensions or become at risk of closure.[14 15] Our findings showed how trials use pilot and/or feasibility studies in an attempt to assess and evaluate prior to a full trial, whether it is likely to be able to recruit its target sample size and whether the study is indeed feasible as a full trial. In both groups, we found pilot studies assessed more study elements than feasibility studies. This also applied to the internal and external studies in group 2;

**Table 2** Group 2: study elements captured in external pilot and feasibility studies

| Study elements | External pilot study (n=48) | | External feasibility study (n=11) | |
| --- | --- | --- | --- | --- |
| | Assessed n (%) | Recommended changes n (%) | Assessed n (%) | Recommended changes (n) |
| Testing recruitment | 24 (50.0) | 3 (6.3) | 8 (72.7) | 0 |
| Determining sample size and/or number available | 24 (50.0) | 1 (2.1) | 4 (36.4) | 1 (9.1) |
| Follow-up/dropout | 16 (33.3) | 0 | 3 (27.3) | 0 |
| Hypothesis testing | 10 (20.8) | 0 | 2 (18.2) | 0 |
| Resources | 2 (4.2) | 0 | 1 (9) | 0 |
| Randomisation | 7 (14.6) | 0 | 3 (27.3) | 0 |
| Blinding | 4 (8.3) | 1 (2.1) | 0 | 0 |
| Outcome measures | 10 (20.8) | 1 (2.1) | 1 (9.1) | 0 |
| Control group | 3 (6.3) | 0 | 0 | 0 |
| Data collection | 6 (12.5) | 0 | 2 (18.2) | 0 |
| Clinical outcomes | 12 (25.0) | 0 | 1 (9.1) | 0 |
| Dose/efficacy/safety | 14 (29.2) | 1 (2.1) | 0 | 0 |
| Acceptability | 17 (35.4) | 0 | 4 (36.4) | 0 |
| Feasibility | 20 (41.7) | 0 | 8 (72.7) | 0 |
| Suggests further study | 8 (16.6) | 1 (2.1) | 1 (9.1) | 0 |
| Delivery of intervention | 8 (16.6) | 2 (4.2) | 0 | 0 |
| Testing/developing materials | 3 (6.3) | 0 | 1 (9.1) | 0 |

**Table 3** Group 2: study elements captured in internal pilot and feasibility studies

| Study elements | Internal pilot study (n=80) Assessed n (%) | Internal feasibility study (n=12) Assessed n (%) |
|---|---|---|
| Testing recruitment | 74 (92.5) | 12 (100.0) |
| Determining sample size and/or number available | 21 (26.3) | 4 (33.3) |
| Follow-up/dropout | 28 (35.0) | 5 (41.7) |
| Hypothesis testing | 0 | 0 |
| Resources | 3 (3.8) | 1 (8.3) |
| Randomisation | 27 (33.8) | 4 (33.3) |
| Blinding | 2 (2.5) | 0 |
| Outcome measures | 16 (20.0) | 2 (16.7) |
| Control group | 0 | 0 |
| Data collection | 21 (26.3) | 2 (16.7) |
| Clinical outcomes | 1 (1.3) | 0 |
| Dose/efficacy/safety | 5 (6.3) | 1 (8.3) |
| Acceptability | 21 (26.3) | 7 (58.3) |
| Feasibility | 35 (43.8) | 11 (91.7) |
| Suggests further study | 0 | 0 |
| Delivery of intervention | 7 (8.8) | 0 |
| Testing/developing materials | 7 (8.8) | 0 |

external and internal pilot studies were used to assess more study elements than feasibility studies.

## Strengths and weaknesses of the study and in relation to other studies

The main strength of the study was the inclusion of all 'HTA funded' studies over a 5-year period. Although the stand-alone group only included 15 studies, this was as expected. For this group, we identified an increase in almost all of the study elements being assessed compared with earlier work by Arain et al.[10] In group 2, over half of the 'HTA funded' applications included a pilot and/or feasibility study (internal and/or external) (161/303). Compared with Arain et al,[10] the findings were similar for the external pilot and feasibility studies cited in terms of the number of study elements assessed and the number of studies included. For example, *testing recruitment* was the most frequently reported element for pilot studies in both the current study and Arain et al, and d*etermining the sample size and the numbers available* was identical in both studies. However, *randomisation, clinical outcomes* and *feasibility* were reported more frequently by Arain et al than the current study. For the external feasibility studies, the current study found more study elements being assessed than that of Arian et al in terms of *testing recruitment, determining the sample size and the numbers available, randomisation, acceptability, feasibility* and *follow-up/dropout*.

For the internal pilot studies, similar findings were found when comparing Arain et al to the current study: d*etermining the sample size and the numbers available, randomisation* and *clinical outcomes* were assessed more in Arain et al than the current study. As with the internal feasibility studies, we found the current study to report more study elements being assessed than that of Arain et al.: *testing recruitment, determining the sample size and the numbers available, follow-up/dropout, randomisation, acceptability* and *feasibility*. These differences, particularly found with the feasibility studies, could be associated with changes over time in the use and understanding of feasibility studies.

This study relied on an adapted version of the Arain et al framework. As some of the study elements were expanded and new ones were added, a direct comparison with Arain et al findings is limited.[10] Given the subjective nature of some of the study elements, we chose to quality assure all data to eradicate and reduce any known errors. Since the analysis was based explicitly on the reporting of the applicants and did not include any subjective account or interpretation of what was reported, we may have under-reported the number of study elements assessed and/or recommended.

We also noted a mismatch in numbers between those assessing study elements and those where recommendations were made in group one. This was due, in part, to how each study element was reported by the applicants. For example, if a study did not specify that they had assessed these elements but made recommendation for changes, we only inferred that they assessed it, but it could not be recorded in the data, hence the mismatch in the findings. This does however highlight the importance of clearly reporting how, what and where the pilot and/or feasibility study had an impact on the design of the definitive trial.

## Implications

The level of appropriateness in the reporting of pilot and feasibility studies could largely be affected by the lack of clarity and awareness of the different study requirements. Despite the growing literature on improving the quality of research to reduce waste in research, there is limited literature pertaining to how pilot and feasibility studies fit into this agenda for change. From what literature there is on pilot and feasibility studies, there is still some confusion about when, why and how it is appropriate to conduct a pilot and/or feasibility study. The findings in this study, even with the use of a well-defined definition by NIHR, still found evidence where applicants did not adhere to the HTA definitions for 'pilot' and 'feasibility' study on research applications. The terminology is still being used interchangeably. Although the commentary on pilot studies by Thabane et al gives a detailed account of the appropriateness of why and how to conduct a pilot study, a comparison with feasibility studies is lacking.[6] It would be helpful to have a more formal distinction between these two terminologies as suggested by Arain et al. A recent study by Eldridge et al goes some way to rectify this by developing a conceptual framework for

defining pilot and feasibility studies.[16] The conceptual framework shows promising results, by being compatible with the MRC guidance on complex interventions,[7] and their descriptor of pilot studies is similar to that of the NIHR definition. However, it is important to note that the Eldridge *et al* conceptual framework is slightly different from that adopted by the NIHR.[16] The clear lack of dichotomy between pilot and feasibility studies is an area for future consideration, not only for funders to encourage more conformity to the published definitions, but for researchers to make better use of the existing literature to better understand the distinction between pilot and feasibility studies.

Having clear definitions of when to use pilot and feasibility studies is important both in terms of their purpose and for clarifying progression to a full trial. However, it is also important to note the limitations of pilot and feasibility studies and when it is not appropriate to conduct this type of study. Pilot and feasibility studies provide valuable information to inform the design of any subsequent definitive study including, for example, approaches to consent, willingness to recruit and randomisation, and adherence to any proposed intervention. Although they are not usually sufficiently powered to provide estimates of effect size, they can provide data that may be useful in helping define the final size of any subsequent study. However, how they are reported, and in what context, requires caution especially when interpreting the findings and extrapolating these to the delivery of a definitive trial.[3 11 12 17]

## Conclusion and recommendations

'HTA funded' research which is inclusive of pilot and feasibility studies is very likely to assess a variety of study elements, which have been evidence-based through this current study using an adapted version of Arain *et al* framework.[10] However, not reviewing the impact of the preliminary work once the trial commences, we have no way of knowing whether the pilot and/or feasibility studies recommendations were instrumental in the successful completion of the trial. If we are able to demonstrate the value of pilot and feasibility studies, we need to place greater emphasis on not only their role in the design stage of a trial but also how this preliminary work contributed favourably, or not, to the completion of the definitive trial. The internal pilot and/or feasibility studies reported in group 2 could be used for the basis of continued work in this area. By following up on this group, we would be able to analyse the successful delivery of the definitive trial and whether the preliminary work had any bearing on this success.

Recommendations include a larger sample of studies across other UK health research funding agencies to determine the frequency and importance of those study elements reported here. A further assessment between the study elements noted in the pilot and feasibility studies and how this impacted on the eventual design and conduct of the definitive trial would certainly add value. This could be achieved by prospectively evaluating the ongoing use of pilot and feasibility studies in group 2 (specifically the internal pilot and/or feasibility studies) as well as future funded applications to the HTA programme. Highlighting the need for better reporting of pilot and feasibility studies should be regarded as relevant to all research funding bodies. And as such, better guidelines for the design, conduct, analysis and reporting of pilot and feasibility studies are still needed.

Future work could therefore include widening the study outcomes presented here to other NIHR funded research programmes. Funders might want to consider the use of Arain *et al* framework when considering the funding of pilot/feasibility studies. Where appropriate this could contribute to maximising the benefit of research and reducing the extent to which research is wasted. If we find ways to appropriately address the flaws detected at the design and conduct stages of research, then we could start to see how research adds value and reduces the amount of research waste. In order to achieve this, we need clearly defined terminology which is inclusive of funding agencies and researchers' perspective; empirical evidence on the reporting and appropriate use of pilot and feasibility studies, in terms of favourable study elements and an evaluation of the contribution to definitive trial outcomes.

**Acknowledgements** The authors acknowledge the NIHR HTA Programme for providing data support during the preliminary stage of the study and the NETSCC Reporting Services team for identifying the relevant applications for the two groups. Professor Jane Blazeby for providing external expert advice during the study.

**Contributors** The study was conceived and designed by MAK, ABJ and EK. The work was undertaken by WP as part of a 4th medical student research project, supervised and supported by MAK, ABJ and EK. Analysis and quality assurance were conducted by WP and MAK and the Access database was prepared by ABJ. All authors read and approved the final manuscript. ABJ is guarantor of the study.

**Funding** This study was supported by the NIHR Evaluation, Trials and Studies Coordinating Centre through the Research on Research programme as part of a University of Southampton, Faculty of Medicine, BM5 Medicine, 4th Year project.

**Competing interests** ABJ and MAK are employed by the University of Southampton to work for NETSCC. ABJ is employed as the Senior Research Fellow for the Research on Research programme and has worked for NETSCC (and its predecessor organisation) in various roles since 2008. MAK is the scientific director at NETSCC and was an editor of the Health Technology Assessment journal. EK is employed by the University of Southampton and works at Southampton's Clinical Trials Unit. EK worked for the Research on Research programme during June 2014 to December 2015. WP was a 4th Year BM5 Medicine student at the University of Southampton.

**Patient consent** Not required.

**Provenance and peer review** Not commissioned; externally peer reviewed.

**Data sharing statement** The datasets used and analysed, and anonymised data are available on request from the corresponding author.

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
