## [Reviewer comments · BMJ Open]

ARTICLE DETAILS

TITLE (PROVISIONAL)	The role of feasibility and pilot studies in randomised controlled trials: a cross-sectional study
AUTHORS	Blatch-Jones, Amanda; Pek, Wei; Kirkpatrick, Emma; Ashton-Key, Martin

VERSION 1 – REVIEW

REVIEWER	Lawrence Mbuagbaw McMaster University
REVIEW RETURNED	23-Feb-2018

GENERAL COMMENTS	The authors investigate the role of pilot and feasibility studies funded by the NIHR HTA program using a retrospective cohort design. I have the following comments. In the strengths and limitations section, this sentence is not clear: “The study contributes to the limited literature in this area whilst maximising the value and importance to adding value research agenda”. Please rephrase. Neither does this one: “The study reviews the different ways pilot and feasibility studies funded by the HTA programme and how they inform the design of a trial (review of study elements)” In the abstract, the conclusion of: “However, caution is required about when it is not appropriate to conduct this type of study” does not stem from any findings in this study. The setting should be the registry/database used to search for studies. Please convert the NETSCC reference into a webpage reference and cite appropriately. In the methods section, it is not clear whether the cohorts were established a priori or the investigators decided to split the data in two after seeing the data. Provide more details on the NETSCC’s research management database and provide a URL. The piloting of the data extraction tools should be reported higher up. Generally, the methods section is hard to follow and doesn’t appear systematic. I would recommend adopting the following chronology:  1. Database searched (terms and limits) 2. Inclusion criteria used. 3. Data management (data retrieval, screening, extraction, quality assurance etc) 4. Data analyses In the data analysis section: “The classification system adapted from Arain et al. was captured using Microsoft Access”. This sentence is not clear. Describe what was entered on the forms. SPSS is not references adequately. It should look something like
--

	this: "IBM Corp. Released 2012. IBM SPSS Statistics for Windows, Version 21.0. Armonk, NY: IBM Corp." There is mention of the use of inferential statistics, but none are reported in the manuscript. The interquartile range is 98 minus 29. You have reported quartiles 3 and 1, not the interquartile range. This is done correctly in the tables, but just the range is needed, not the arithmetics behind it. A description of the kinds of studies include here would be helpful as HTA is a broad field. In the figures, please add the reasons why the studies were excluded. The contribution of this paper to the literature is not very clear. There is no rationale for assuming all HTA studies would want to pilot test all study elements in the Arain framework.
--	---

REVIEWER	Steven A. Julious The University of Sheffield, England
REVIEW RETURNED	03-Mar-2018

GENERAL COMMENTS	Overall Comments This is a kind of paper I like as reviewing evidence of trials is a way one can learn from case law both good and bad. I have a comments made to clarify and (hopefully) enhance the paper Major Comments  1. The paper is not a cohort study. What has been undertaken is an audit. The title of the paper and references to cohorts within the paper and the study design should be amended to reflect this 2. Reference is made to the NETSCC research management database with no reference. Is this a publicly available database researchers can access? If so then it would be good to state the procedure. If not then this needs to be stated 3. Would the database search include studies funded by RfPB as many of these would be pilot studies. 4. It needs to be highlighted in the paper more that outside of NETSCC few people see a dichotomy between pilot and feasibility studies (p12). The authors cite the paper Eldridge who discuss the Delphi and workshop they undertook. Due to the push back on mutual exclusive definitions of pilot and feasibility studies they came up with a framework to reflect practice. The fact that NETSCC funded studies do not conform with NETSCC definitions of pilot and feasibility studies is in itself very interesting. Outside of NETSCC funded studies one could content the differences would be event less clear 5. Given the affiliations of the authors I am surprised that they have not cited of the oeuvre of Amy Whitehead also at Southampton on pilot studies the most pertinent would be Cooper CL, Whitehead A, Potrill E, Julious SA and Walters SJ. Are pilot trials useful for predicting randomisation and attrition rates in definitive studies: A review of publicly funded trials. Clinical Trials 2018: DOI: 10.1177/1740774517752113 Whitehead AL, Julious SA, Cooper CL and Campbell MJ. Estimating the sample size for a pilot randomised trial to minimise the overall trial sample size for the external pilot and main trial for a continuous
---

outcome variable. *Statistical Methods in Medical Research* 2016 25(3) 1057-1073 (DOI: 10.1177/0962280215588241)
Billingham SAM, Whitehead AL and Julious SA. An audit of sample sizes for pilot and feasibility trials being undertaken in the United Kingdom registered in the United Kingdom Clinical Research Network database. *BMC Medical Research Methodology* 2013, 13:104 doi:10.1186/1471-2288-13-104

6. Table 1 in Billingham needs to be produced in the paper for each data set to give some background to the trials: what were being evaluated (drugs or health technologies); how many arms are in the studies (sample size is given but not clear if sample size per arm or total sample size the former is preferred); disease areas and type of endpoint (which would influence the sample size). Also, it is also not clear if these trials are individually randomised

7. Also similar to Billingham more details need to be put into Figure 1 and 2 on exclusions etc

8. Cooper et al is quite pertinent to this paper as pairs of studies (pilot and main trials) were analysed from the HTA. Pilots were show not to be too great at predicting recruitment and dropouts in the main trial (which is not surprising as if the pilot was poor one would hope either the main would not take place or remedial action would be taken in the main trial learning from the pilot)

9. It should be highlighted more that data set 1 and data set 2 are extracted from different sources. Are any of data set 1 quoted in data set 2?

10. For the main trials citing pilot studies were these external pilot studies evaluated? From experience these might not be actual pilots in the sense as defined in this paper. For example the authors may have reviewed medical notes to see how many patients would be eligible for the study and then call this a pilot or feasibility study

11. The conclusions and recommendations at the end of the paper could be improved. Most of the text seems to be talking about following up the data or planning to follow up the data sets or (if followed up it could be a cohort). Recommendations should be more along the lines of what you would recommend from what you have seen to improve things.

12. It should be given as a weakness that this work is from one funding stream in one country (especially given none NETSCC would likely be different in pilot and feasibility trial definitions)

Other comments

13. Could the data sets be posted on line with the publications as opposed to being available upon request

14. The description of Table 1 should be expanded. It would not be immediately clear to some readers what these rows are and so should be described. Did the authors have a data extraction form?

15. The labels in Table 2, 3 and 4 need improving. Is "Determining SS" assessing if the authors were from 1930s Germany? The abbreviations RC and A do not add much. What is

	meant by control group? These tables do tie in with Table 1 but the tables could be clearer. The fact that hypothesis testing was done in external pilots should be highlighted (as Arain amongst others says this should not be done if not set out to hypothesis test) 16. Figure 3 and Figure 4 do not seem to be adding too much to the paper especially as all the data are in the Tables. If to include should not have number of participants and median as columns 17. Can all web links have date last accessed
--	---

REVIEWER	Julius Sim Keele University, UK
REVIEW RETURNED	07-Mar-2018

GENERAL COMMENTS	p4, l12: the term 'pilot/feasibility studies' is a little unclear; does the '/' mean 'and', 'or', or 'and/or'? Applies elsewhere. p5, l12: in what sense is poor research design 'associated with' the factors? Do you mean something like 'includes'? p5, l16: it is not clear what distinction is being made between 'poor' and 'inadequate' methods. p6, l9: maybe this would be better described as an objective rather than a purpose. p7, l15: how would the title of the journal indicate the eligibility of a study? Do you mean the title of the journal article? p7, l17: I don't follow the phrase 'to confirm the citation of or main study type as'. p8, l41: I cannot find any inferential analyses in this report - they seem all to be descriptive. p9, l47: it would be helpful to name the five categories here, in addition to presenting them in Figure 2. p9, l49: it's not clear what is being referred to as 'citations'. How do they differ from the 'originally cited reference'? p10, l0: in what sense are these 'criteria'? They sound more like objectives. Also applies to tables 2-4. p12, l35: In what sense should pilot or feasibility studies not perform sample size calculations? They may be used to estimate parameters that are then used in sample size calculations. Can you be more specific as to the role that you believe they should or should not perform in relation to sample size calculations? Table 2: presumably 'SS' stands for sample size. Do you mean 'and' (i.e. that both the sample size and the number of available participants were assessed) or do you mean 'and/or'? Table 2: in the final row, what does 'being equal to' mean? Minor points:
--

	p2, l 34: change 'was conducted' to 'were conducted'. p4, l4 and p5, l3. Initial capitals not needed for 'randomised controlled trials' p4, l4: abbreviation RCT needs to be defined here (insert '(RCTs)' after 'randomised controlled trials') p4, l20: perhaps omit 'and determine'. p4, l33: change 'assess' to 'assesses'. p5, l49: change 'for when' to 'of when'. p7, l22: there is no main verb in this sentence. p7, l31: there is no main verb in this sentence. p9, l5: 'HTA funded' should be hyphenated (and elsewhere). p9, l7: do you mean 'identified as eligible'? p9, l53: delete 'of' after 'comprised'. p10, l37: I think you need a comma, not a colon, after 'changes'. p11, l14: I don't understand what 'that of' refers to. p12, l15: the 'we' does not fit here. Reword this sentence. p12, l26: change 'different to' to 'different from'. p12, l46: change 'are' to 'is'. p12, l52: why 'predicted'? Do you mean 'contributed'? p13, l18: There is no main verb in this sentence.
--	---

REVIEWER	Mike Campbell University of Sheffield UK
REVIEW RETURNED	11-Mar-2018

GENERAL COMMENTS	This paper is an observational study of 15 stand-alone pilot/feasibility studies and 161 funded trials. The authors have a unique opportunity to review studies funded by the NIHR for the way pilot/feasibility studies are reported and used by NIHR funded studies . To some extent it repeats the work of Arain et al (2010) looking at the differences between studies described as 'pilot' and those as 'feasibility' Comments 1) The NIHR has not been long established and so the time period covered is only 5 years. With only 15 stand alone studies it is difficult to know how generalizable the findings are. Thus I am not sure that Table 2 contributes much. 2) The authors could reference the new CONSORT guidelines for pilot/feasibility studies (1), and comment to what extent the studies reported in their study adhered to these guidelines. 3) The authors may care to liaise with Dr Ben Morgan of the NIHR in
---

	London, who has carried out a similar review, as yet unpublished Minor 1) Not really clear why two percentages were quoted for , say, testing recruitment , in the Abstract. I assume they relate to the 161 full applications or the 59 that cited a pilot/feasibility study? 2) P9 'comprised' not 'Comprised of' Ref 1 Eldridge SM, Chan CL, Campbell MJ, Bond CM, Hopewell S, Thabane L, Lancaster GA on behalf of the PAFS consensus group. CONSORT 2010 statement: extension to randomised pilot and feasibility trials BMJ 2016;355:i5239
--	---

VERSION 1 – AUTHOR RESPONSE

Reviewer 1 comments	Authors' responses
In the strengths and limitations section, this sentence is not clear: "The study contributes to the limited literature in this area whilst maximising the value and importance to adding value research agenda". Please rephrase.	This has been amended accordingly – see main manuscript
Neither does this one: "The study reviews the different ways pilot and feasibility studies funded by the HTA programme and how they inform the design of a trial (review of study elements)"	This has been amended accordingly – see main manuscript
In the abstract, the conclusion of: "However, caution is required about when it is not appropriate to conduct this type of study" does not stem from any findings in this study.	The conclusions under 'implications' clearly discuss when and how it is appropriate to conduct a pilot and/or feasibility study. Thabane et al and Arain et al studies were used as references. The conclusions in the abstract have therefore not been changed although the text has been closely aligned to what is reported in the conclusions.
The setting should be the registry/database used to search for studies.	I have kept to the standard approach used in previous submissions and reviewed other similar published papers in BMJ Open.
Please convert the NETSCC reference into a webpage reference and cite appropriately.	This has been amended
In the methods section, it is not clear whether the cohorts were established a priori or the investigators decided to split the data in two after seeing the data.	An explanation of the data and cohorts have been included under 'data source' to further explain the cohorts
Provide more details on the NETSCC's research management database and provide a URL.	There is no URL for NETSCC's research management database – this is the system used to manage the HTA (and other programmes of work) programme. I have amended the text to be consistent with other published work using the same source.
The piloting of the data extraction tools should be reported higher up.	I have moved the 'piloting' section above 'classification systems' sub-heading.
Generally, the methods section is hard to follow and doesn't appear systematic. I would	I have made some slight changes to the structure of the methods section. However,

recommend adopting the following chronology:  1. Database searched (terms and limits) 2. Inclusion criteria used. 3. Data management (data retrieval, screening, extraction, quality assurance etc) 4. Data analyses 	keeping in line with the guidelines and previously published work published in BMJ Open and other journals of similar work the authors feel the ordering of the methods section is adequate. As this piece of work is not a systematic review, which would of course follow the suggested sub headings, these headings are not appropriate for the purposes of the current study.
In the data analysis section: "The classification system adapted from Arain et al. was captured using Microsoft Access". This sentence is not clear. Describe what was entered on the forms.	The sentence has been amended and includes more details about the process we did in Access. I have also referenced Access.
SPSS is not references adequately. It should look something like this: "IBM Corp. Released 2012. IBM SPSS Statistics for Windows, Version 21.0. Armonk, NY: IBM Corp."	I have amended the reference to SPSS to the standard formatting used in previous publications.
There is mention of the use of inferential statistics, but none are reported in the manuscript. The interquartile range is 98 minus 29. You have reported quartiles 3 and 1, not the interquartile range. This is done correctly in the tables, but just the range is needed, not the arithmetics behind it.	This has been changed. The range is also included in Table 2.
A description of the kinds of studies include here would be helpful as HTA is a broad field.	The type of studies included are reported throughout the paper and are clearly identified in the methods and results section. Cohort 1 – Standalone pilot and feasibility studies funded by the HTA programme Cohort 2 – RCTs funded by the HTA programme If the titles of the included studies are required, an appendix could be provided.
In the figures, please add the reasons why the studies were excluded.	These have been included in the figures
The contribution of this paper to the literature is not very clear.	The purpose and objectives of the paper are outlined at the end of the introduction. The authors have explained why Arain et al framework is being used and how it will help inform how pilot/feasibility studies are used to inform full RCTs
There is no rationale for assuming all HTA studies would want to pilot test all study elements in the Arain framework.	I agree and the conclusions do not suggest all HTA studies need a pilot/feasibility, consideration is needed as to the suitability and uncertainty of that particular study on a case by case basis.

Reviewer 2 comments	Authors' responses
The paper is not a cohort study. What has been undertaken is an audit. The title of the paper and references to cohorts within the paper and the study design should be amended to reflect this	After discussion and reviewing previously published work in a similar field from the same research group at NETSCC (Health Research Policy and Systems201513:37

	https://doi.org/10.1186/s12961-015-0025-8) The authors concur that this is a retrospective account of published and active research funded by the HTA programme and includes two cohorts. Both cohorts have been assessed and evaluated using Arain et al framework to determine which elements are exposed in each of the included studies.
Reference is made to the NETSCC research management database with no reference. Is this a publicly available database researchers can access? If so then it would be good to state the procedure. If not then this needs to be stated	Based on previous reviewer comments the reference to the NETSCC database has been amended. It is not a public available system as it is the research management system for NETSCC to manage the contract of the research programmes.
Would the database search include studies funded by RfPB as many of these would be pilot studies.	The NETSCC research management system does not include RfPB as this is centrally managed by a different coordinating centre.
It needs to be highlighted in the paper more that outside of NETSCC few people see a dichotomy between pilot and feasibility studies (p12). The authors cite the paper Eldridge who discuss the Delphi and workshop they undertook. Due to the push back on mutual exclusive definitions of pilot and feasibility studies they came up with a framework to reflect practice. The fact that NETSCC funded studies do not conform with NETSCC definitions of pilot and feasibility studies is in itself very interesting. Outside of NETSCC funded studies one could content the differences would be event less clear	Thank you for your comment and suggestion. I have extended the 'implications' section to better explain the lack of separation between pilot and feasibility studies.
Given the affiliations of the authors I am surprised that they have not cited of the oeuvre of Amy Whitehead also at Southampton on pilot studies the most pertinent would be Cooper CL, Whitehead A, Potrill E, Julious SA and Walters SJ. Are pilot trials useful for predicting randomisation and attrition rates in definitive studies: A review of publicly funded trials. Clinical Trials 2018: DOI: 10.1177/1740774517752113 Whitehead AL, Julious SA, Cooper CL and Campbell MJ. Estimating the sample size for a pilot randomised trial to minimise the overall trial sample size for the external pilot and main trial for a continuous outcome variable. Statistical Methods in Medical Research 2016 25(3) 1057-1073 (DOI: 10.1177/0962280215588241) Billingham SAM, Whitehead AL and Julious SA. An audit of sample sizes for pilot and feasibility trials being undertaken in the United Kingdom registered in the United Kingdom Clinical Research Network database. BMC Medical Research Methodology 2013, 13:104 doi:10.1186/1471-2288-13-104	Thank you. The publication of Whitehead et al. work was at the time of submission of the current paper. Given the opportunity to re-submit we have included the work by Whitehead et al into the revised paper and other noted published articles. Thank you
Table 1 in Billingham needs to be produced in the paper for each data set to give some background to the trials: what were being	The data reported in Table 1 of the Billingham et al paper was not extracted for the purposes of the current study.

evaluated (drugs or health technologies); how many arms are in the studies (sample size is given but not clear if sample size per arm or total sample size the former is preferred); disease areas and type of endpoint (which would influence the sample size). Also, it is also not clear if these trials are individually randomised	The characteristics of the pilot and feasibility studies were not part of the objectives of the study and including such data would not add anything to the paper. The purpose of the paper was to determine what study design elements were included or referenced. Following the Arain et al paper we were examining the methodological components of the study design rather than detailing what these were. Going back to the included studies is unfortunately out of scope. Data reporting whether the trials were individual or cluster randomised was not extracted.
Also similar to Billingham more details need to be put into Figure 1 and 2 on exclusions etc	Exclusions have been included in Figure 1 and Figure 2 and text has been included into the manuscript.
Cooper et al is quite pertinent to this paper as pairs of studies (pilot and main trials) were analysed from the HTA. Pilots were show not to be too great at predicting recruitment and dropouts in the main trial (which is not surprising as if the pilot was poor one would hope either the main would not take place or remedial action would be taken in the main trial learning from the pilot)	As above – paper now referenced in the manuscript
It should be highlighted more that data set 1 and data set 2 are extracted from different sources. Are any of data set 1 quoted in data set 2?	Additional information has been included in the methods section to clearly state the difference sources used for Cohort 1 and Cohort 2. The reporting of data for each cohort is separate, hence the reporting style under the sub-headings ‘cohort 1’ and ‘cohort 2’. No data from either cohort is reported under the other cohort.
For the main trials citing pilot studies were these external pilot studies evaluated? From experience these might not be actual pilots in the sense as defined in this paper. For example the authors may have reviewed medical notes to see how many patients would be eligible for the study and then call this a pilot or feasibility study	The manuscript states “For the 59 applications where an external pilot / feasibility study was referenced, we found that not all of these studies provided information relating to the number of participants that took part in the pilot/feasibility study. We did not go back to the original journal article to retrieve this information. Therefore, it was not appropriate to estimate the median or inter quartile range for this cohort.” The authors agree that this answers the reviewers comment.
The conclusions and recommendations at the end of the paper could be improved. Most of the text seems to be talking about following up the data or planning to follow up the data sets or (if followed up it could be a cohort). Recommendations should be more along the lines of what you would recommend from what you have seen to improve things.	A number of recommendations have been suggested in the conclusions relating to the continued reporting of the checklist, recommendations for consideration by funders and areas for consideration by researchers. Implications for future research is also included in the conclusions

It should be given as a weakness that this work is from one funding stream in one country (especially given none NETSCC would likely be different in pilot and feasibility trial definitions)	This has been included as a limitation
Other comments:	
Could the data sets be posted on line with the publications as opposed to being available upon request	We have applied the same criteria as other published work from NETSCC RoR. A statement has been included under the 'Data Sharing' subheading. "The datasets used and analysed, and anonymised data are available on request from the corresponding author." This is in line with previous publications "The datasets used and analysed during the current study are available on request from the Research on Research team: ror@nhr.ac.uk." AND "Anonymised data may be requested from the corresponding author."
The description of Table 1 should be expanded. It would not be immediately clear to some readers what these rows are and so should be described. Did the authors have a data extraction form?	Additional information related to the content of Table 1 has been included. As we used the methodological components reported by Arain et al, we followed the same process that was published in the journal article. Yes, as explained under Methods we used Access to develop forms for data extraction for each included study.
The labels in Table 2, 3 and 4 need improving. Is "Determining SS" assessing if the authors were from 1930s Germany? The abbreviations RC and A do not add much. What is meant by control group? These tables do tie in with Table 1 but the tables could be clearer. The fact that hypothesis testing was done in external pilots should be highlighted (as Arain amongst other says this should not be done if not set out to hypothesis test)	"Determining SS" has been changed to Determining the Sample Size in all relevant tables. The A and RC abbreviations are used in Table 2 as the third column in each set (pilot studies, feasibility studies and pilot/feasibility studies) shows the number of studies that reported both the assessment of the study element and recommended changes to the study element. This cannot be determined by merely looking at the first two columns in Table 2. No changes have been made in relation to this comment. Control group refers to whether the pilot or feasibility study were identified as using a control group (as per stated by Arain et al publication). Hypothesis testing results were small by comparison to the other study elements. The tables show this and due to limited word count, it is not possible for the authors to discuss all of the study elements, in turn, in the paper. Therefore, only the main findings from the study elements were discussed in more detail.
Figure 3 and Figure 4 do not seem to be adding too much to the paper especially as all the data are in the Tables. If to include should not have number of participants and median as columns	The authors agree to take out Figures 3 and 4. The number of participants (median and range) has been amended based on other reviewer comments. Changes to Table 2 last row (Median number of participants) does not detect

	track changes. The columns have been deleted and the row is presented by merging the three associated columns.
Can all web links have date last accessed	This is included

Reviewer 3 comments	Authors' responses
p4, l12: the term 'pilot/feasibility studies' is a little unclear; does the '/' mean 'and', 'or', or 'and/or'? Applies elsewhere.	Have rephrased where appropriate to make the distinction clearer for the reader
p5, l12: in what sense is poor research design 'associated with' the factors? Do you mean something like 'includes'?	Sentence rephrased: “Poorly designed trials could include non-reference to a pre-existing systematic literature review or bias generated by inadequate concealment of treatment allocation. ¹ Research by Cooper et al. has also shown variability between external pilots and the prediction for randomisation and attrition rates. ³ As a result, much attention has primarily focused on the design, conduct and analysis of clinical research to determine where improvements are needed to reduce waste in research.”
p5, l16: it is not clear what distinction is being made between 'poor' and 'inadequate' methods.	See above
p6, l9: maybe this would be better described as an objective rather than a purpose.	Thank you – I have changed the manuscript
p7, l15: how would the title of the journal indicate the eligibility of a study? Do you mean the title of the journal article?	Yes, thank you
p7, l17: I don't follow the phrase 'to confirm the citation of or main study type as'.	I have re-written the section to better explain this
p8, l41: I cannot find any inferential analyses in this report - they seem all to be descriptive.	Yes, you are correct. Text amended to reflect this.
p9, l47: it would be helpful to name the five categories here, in addition to presenting them in Figure 2.	Thank you, I have added these to the main manuscript
p9, l49: it's not clear what is being referred to as 'citations'. How do they differ from the 'originally cited reference'?	I have rephrased to “For the 59 applications where an external pilot / feasibility study was referenced, we found that not all of these studies provided information relating to the number of participants that took part in the pilot/feasibility study. As such we did not go back to the original journal article to retrieve this information. Therefore, it was not appropriate to estimate the median or inter quartile range for this cohort.”
p10, l0: in what sense are these 'criteria'? They	The text is referring to the study elements so I

sound more like objectives. Also applies to tables 2-4.	have deleted 'criteria assessed in...' to make more sense.
p12, l35: In what sense should pilot or feasibility studies not perform sample size calculations? They may be used to estimate parameters that are then used in sample size calculations. Can you be more specific as to the role that you believe they should or should not perform in relation to sample size calculations?	I have rephrased to "Pilot and feasibility studies are to assist and direct the design stage of a trial, they should not be used to assess effectiveness, make claims about whether the treatment works or not, or perform sample size calculations to produce effect size estimates for a larger trial. Feasibility studies are not adequately powered to assess effectiveness; the sample sizes are too small to give a true effect size estimates."
Table 2: presumably 'SS' stands for sample size. Do you mean 'and' (i.e. that both the sample size and the number of available participants were assessed) or do you mean 'and/or'?	Yes SS refers to sample size. Text changed in the relevant tables. Yes, we do mean and/or. I have changed the n to number.
Table 2: in the final row, what does 'being equal to' mean?	This has been deleted from the tables
Minor points:	
p2, l 34: change 'was conducted' to 'were conducted'.	Manuscript changed
p4, l4 and p5, l3. Initial capitals not needed for 'randomised controlled trials'	Manuscript changed
p4, l4: abbreviation RCT needs to be defined here (insert '(RCTs)' after 'randomised controlled trials')	Manuscript changed
p4, l20: perhaps omit 'and determine'.	Manuscript changed
p4, l33: change 'assess' to 'assesses'.	Manuscript changed
p5, l49: change 'for when' to 'of when'.	Manuscript changed
p7, l22: there is no main verb in this sentence.	Manuscript changed
p7, l31: there is no main verb in this sentence.	Manuscript changed
p9, l5: 'HTA funded' should be hyphenated (and elsewhere).	Manuscript changed
p9, l7: do you mean 'identified as eligible'?	Manuscript changed
p9, l53: delete 'of' after 'comprised'.	Manuscript changed
p10, l37: I think you need a comma, not a colon, after 'changes'.	Manuscript changed
p11, l14: I don't understand what 'that of' refers to.	'that of' deleted
p12, l15: the 'we' does not fit here. Reword this sentence.	'we' deleted
p12, l26: change 'different to' to 'different from'.	Manuscript changed
p12, l46: change 'are' to 'is'.	Manuscript changed

p12, 152: why 'predicted'? Do you mean 'contributed'?	Yes, thank you manuscript changed
p13, 118: There is no main verb in this sentence.	Manuscript changed

Reviewer 4 comments	Authors' responses
The NIHR has not been long established and so the time period covered is only 5 years. With only 15 standalone studies it is difficult to know how generalizable the findings are. Thus I am not sure that Table 2 contributes much.	Having found only 15 standalone pilot/feasibility studies funded by the HTA programme is a finding in itself. The findings are reporting only the HTA programme and gives us some indication for future follow up (across other programmes at a later date). This is discussed in the discussion and conclusion.
The authors could reference the new CONSORT guidelines for pilot/feasibility studies (1), and comment to what extent the studies reported in their study adhered to these guidelines.	Thank you, we have included and referenced the new CONSORT guidelines for pilot/feasibility studies
Minor 1)Not really clear why two percentages were quoted for , say, testing recruitment , in the Abstract. I assume they relate to the 161 full applications or the 59 that cited a pilot/feasibility study?	Abstract changed to reflect a clearer understanding. The two percentages relate to either pilot or feasibility study. For cohort 2 there were also internal and external pilot or feasibility studies, making four groups. 59 were external and 92 were internal (totally 161 in cohort 2).
P9 'comprised' not 'Comprised of'	Amended (as per other comment)
Ref 1 Eldridge SM, Chan CL, Campbell MJ, Bond CM, Hopewell S, Thabane L, Lancaster GA on behalf of the PAFS consensus group. CONSORT 2010 statement: extension to randomised pilot and feasibility trials BMJ 2016;355:i5239	As above – reference included in the revised manuscript

VERSION 2 – REVIEW

REVIEWER	Lawrence Mbuagbaw McMaster University
REVIEW RETURNED	17-Apr-2018

GENERAL COMMENTS	Most of the comments I raised have been addressed.
--

REVIEWER	Steven Julious The University of Sheffield
REVIEW RETURNED	03-May-2018

GENERAL COMMENTS	Comments not address from previously Point 1: "The paper is not a cohort study. What has been undertaken is an audit. The title of the paper and references to cohorts within the paper and the study"
---

	It is not a cohort study. Reference to other papers which may have got the definition wrong (I did not read them in detail) does not mean in the current research should be consistent. Personally I would call the work an audit. It called also be called a cross sectional survey as the authors prefer As mentioned in my previous feedback was if the plan was to follow these data up then that would make the data a cohort but it would need to be explicitly stated that this was the first analysis of a planned follow up Point 2: "Table 1 in Billingham needs to be produced in the paper for each data set to give some background to the trials: what were being evaluated (drugs or health technologies); how many arms are in the studies (sample size is given but not clear if sample size per arm or total sample size the former is preferred); disease areas and type of endpoint (which would influence the sample size). Also, it is also not clear if these trials are individually randomised" Where this is important is that by extracting some basic information it informs the reader if the work can be generalised beyond the NETSCC databse - for example do the trials look similar to trials one is undertaking? In this paper the authors the average sample size. The sample size is influenced by both the number of treatment arms and whether the study is cluster randomised.
--	--

REVIEWER	Julius Sim Keele University, UK
REVIEW RETURNED	10-May-2018

GENERAL COMMENTS	Many thanks for the changes that you have made in response to my earlier comments. There is just one issue on which further clarification is required, I think. I suggested that you should indicate why you believe pilot and feasibility studies should not be used to 'perform sample size calculations'. The revision to the paper reads '...should not be used to assess effectiveness, make claims about whether the treatment works or not, or perform sample size calculations to produce effect size estimates for a larger trial. Feasibility studies are not adequately powered to assess effectiveness; the sample sizes are too small to give true effect size estimates'. The question of assessing effectiveness and that of estimating parameters for a sample size calculation need to be addressed separately, whereas they seem to be rather confounded here. For the sample size objective, the concern is not normally with providing measures of effectiveness or estimating 'effect sizes.' An effect size for a sample size calculation requires a pre-specified meaningful difference, which is not normally determined empirically from data, but is based on clinical and/or theoretical considerations (note also that any effect size is not 'produced' by the calculation, but is part of the input to the calculation). Instead, the focus is usually on estimating key components of the sample size calculation, such as the variance of a continuous outcome, the distribution of the data, the consent rate, the rate of missing values on measurement tools, and the rate of loss to follow-up. There are several papers in the literature that suggest that these
---

	can be estimated (or, in the case of distributions, assessed) from pilot or feasibility studies of an appropriate size (at least in respect of individually randomized trials - some special considerations may apply in pilot cluster trials). Can you explain more fully why you seem to regard this as misguided?
REVIEWER	Mike Campbell ScHARR, University of Sheffield Uk The article quotes papers I have written and so its publication would boost my number of citations
REVIEW RETURNED	27-Apr-2018
GENERAL COMMENTS	The authors have responded to my comments adequately.

VERSION 2 – AUTHOR RESPONSE

Reviewer 1	Authors response
Point 1: "The paper is not a cohort study. What has been undertaken is an audit. The title of the paper and references to cohorts within the paper and the study" It is not a cohort study. Reference to other papers which may have got the definition wrong (I did not read them in detail) does not mean in the current research should be consistent. Personally I would call the work an audit. It called also be called a cross sectional survey as the authors prefer	Thank you. We have changed to cross-sectional study. We have amended text throughout the paper to reflect this.
Point 2: "Table 1 in Billingham needs to be produced in the paper for each data set to give some background to the trials: what were being evaluated (drugs or health technologies); how many arms are in the studies (sample size is given but not clear if sample size per arm or total sample size the former is preferred); disease areas and type of endpoint (which would influence the sample size). Also, it is also not clear if these trials are individually randomised" Where this is important is that by extracting some basic information it informs the reader if the work can be generalised beyond the NETSCC databse - for example do the trials look similar to trials one is undertaking? In this paper the authors the average sample size. The sample size is influenced by both the number of treatment arms and whether the study is cluster randomised.	Thank you for your comment. Following the analysis conducted in the Arain et al paper, we did not therefore collect trial characteristic data. We did not calculate the sample sizes as part of the extraction process. The characteristics of the trials were not extracted and did not form part of the analysis. We only captured what was reported to us in the application. We did not retrieve the publications cited and it was never the intention to do so. In order for us to go back through the data we would need extended time for completion which is unfortunately out of scope. The authors appreciate the value of the data and have included the generalisability as a limitation to the paper.
Reviewer 3	
There is just one issue on which further clarification is required, I think. I suggested that you should indicate why you believe pilot and feasibility studies should not be used to 'perform sample size calculations'. The revision to the paper reads '...should not be used to assess effectiveness, make claims about whether the treatment works or not, or perform sample size calculations to produce effect size estimates for a larger trial. Feasibility studies are not adequately powered to assess	Thank you for your additional and useful comments on this point. We have amended the relevant text to read as "Having clear definitions of when to use pilot and feasibility studies is important both in terms of their purpose and for clarifying progression to a full trial. However, it is also important to note the limitations of pilot and feasibility studies

effectiveness; the sample sizes are too small to give true effect size estimates'. The question of assessing effectiveness and that of estimating parameters for a sample size calculation need to be addressed separately, whereas they seem to be rather confounded here. For the sample size objective, the concern is not normally with providing measures of effectiveness or estimating 'effect sizes.' An effect size for a sample size calculation requires a pre-specified meaningful difference, which is not normally determined empirically from data, but is based on clinical and/or theoretical considerations (note also that any effect size is not 'produced' by the calculation, but is part of the input to the calculation). Instead, the focus is usually on estimating key components of the sample size calculation, such as the variance of a continuous outcome, the distribution of the data, the consent rate, the rate of missing values on measurement tools, and the rate of loss to follow-up. There are several papers in the literature that suggest that these can be estimated (or, in the case of distributions, assessed) from pilot or feasibility studies of an appropriate size (at least in respect of individually randomized trials - some special considerations may apply in pilot cluster trials). Can you explain more fully why you seem to regard this as misguided?	and when it is not appropriate to conduct this type of study. Pilot and feasibility studies provide valuable information to inform the design of any subsequent definitive study including for example, approaches to consent, willingness to recruit and randomisation, and adherence to any proposed intervention. Although they are not usually sufficiently powered to provide estimates of effect size they can provide data that may be useful in helping define the final size of any subsequent study. However, how they are reported, and in what context, requires caution especially when interpreting the findings and extrapolating these to the delivery of a definitive trial.”
--	---

VERSION 3 – REVIEW

REVIEWER	Julius Sim Keele University, UK
REVIEW RETURNED	08-Jul-2018
GENERAL COMMENTS	Thank you for responding to my comment regarding sample size calculation.